# A Specific Pattern of Routine Cerebrospinal Fluid Parameters Might Help to Identify Cases of West Nile Virus Neuroinvasive Disease

**DOI:** 10.3390/v16030341

**Published:** 2024-02-22

**Authors:** Johann Otto Pelz, Christoph Mühlberg, Isabel Friedrich, Lorenz Weidhase, Silke Zimmermann, Melanie Maier, Corinna Pietsch

**Affiliations:** 1Department of Neurology, University Hospital Leipzig, 04103 Leipzig, Germany; 2Medical Intensive Care Unit, University Hospital Leipzig, 04103 Leipzig, Germany; 3Institute of Laboratory Medicine, Clinical Chemistry and Molecular Diagnostics, University Hospital Leipzig, 04103 Leipzig, Germany; 4Department of Virology, Institute of Medical Microbiology and Virology, University Hospital Leipzig, 04103 Leipzig, Germany

**Keywords:** West Nile virus, meningitis, encephalitis, cerebrospinal fluid, climate change, immunoglobulin M

## Abstract

Background: Viral meningitis/encephalitis (ME) is a rare but potentially harmful disease. The prompt identification of the respective virus is important to guide not only treatment but also potential public health countermeasures. However, in about 40% of cases, no virus is identified despite an extensive diagnostic workup. The aim of the present study was to analyze demographic, seasonal, and routine cerebrospinal fluid (CSF) parameters in cases of viral ME and assess their utility for the prediction of the causative virus. Methods: Demographic data, season, and routine CSF parameters (total leucocytes, CSF cell differentiation, age-adjusted CSF/serum albumin ratio, and total immunoglobulin ratios) were retrospectively assessed in cases of viral ME. Results: In total, 156 cases of acute viral ME (74 female, median age 40.0 years) were treated at a tertiary-care hospital in Germany. Specific viral infections were detected in 93 (59.6%) cases. Of these, 14 (9.0%) cases were caused by herpes simplex virus (HSV), 36 (23.1%) by varicella-zoster virus (VZV), 27 (17.3%) by enteroviruses, 9 (5.8%) by West Nile virus (WNV), and 7 (4.5%) by other specific viruses. Additionally, 64 (41.0%) cases of ME of unknown viral etiology were diagnosed. Cases of WNV ME were older, predominantly male, showed a severe disruption of the blood–CSF–barrier, a high proportion of neutrophils in CSF, and an intrathecal total immunoglobulin M synthesis in the first CSF sample. In a multinominal logistic regression analysis, the accuracy of these CSF parameters together with age and seasonality was best for the prediction of WNV (87.5%), followed by unknown viral etiology (66.7%), VZV (61.8%), and enteroviruses (51.9%). Conclusions: Cases with WNV ME showed a specific pattern of routine CSF parameters and demographic data that allowed for their identification with good accuracy. These findings might help to guide the diagnostic workup in cases with viral ME, in particular allowing the timely identification of cases with ME due to WNV.

## 1. Introduction

Infections of the central nervous system, i.e., a meningitis/encephalitis (ME), are rare but potentially harmful diseases. Etiologies in immunocompetent patients comprise predominantly viral and bacterial pathogens [1]. In a multicenter prospective observational cohort study of adults with suspected meningitis in the United Kingdom, most cases (42%) had an unknown cause, 36% of cases were viral with an annual incidence for viral meningitis of 2.73 per 100,000, and 16% of cases had a bacterial cause [2]. In general, the timely identification of the respective pathogen is of utmost importance to guide the appropriate treatment [1]. Focusing on viral ME, without treatment or in a case of delayed and/or insufficient treatment, mortality may be dramatically high, like in herpes simplex virus (HSV) encephalitis [3]. Moreover, the spectrum of viruses causing ME is numerous and may vary substantially between global regions [4]. This is even more important in times of climate change when autochthonous transmissions of formerly absent viruses like West Nile virus (WNV) are emerging in Germany [5,6]. At the same time, the area at risk for transmission of tick-borne encephalitis (TBE) is continuously spreading northwards in Germany and Europe [7,8].

During a WNV outbreak in Saxony in 2020 [6], an intrathecal immunoglobulin M synthesis in the Reiber diagram [9], already determined in the initial examination of the cerebrospinal fluid (CSF), and a neutrophilic pleocytosis were observed in several cases of WNV neuroinvasive disease (WNND). Although the early detection of WNV-specific immunoglobulin M in the CSF has repeatedly been described in patients with WNND [10,11,12], it is unclear whether the sole detection of an increased intrathecal IgM synthesis might serve as a hint for the specific WNV-IgM. Noteworthily, hitherto, an intrathecal immunoglobulin synthesis was considered to be unlikely in the early stage of viral ME [1]. A neutrophilic pleocytosis, which is indicative of bacterial meningitis [13], was found not only in WNND [14,15,16] but also in ME caused by enterovirus [17].

Both observations gave reason to review routine CSF parameters from cases with viral ME and assess whether there are patterns that are predictive for specific viruses, in particular for WNV.

## 2. Methods

This retrospective, non-interventional, explorative study was performed according to the ethical standards laid down in the 1964 Declaration of Helsinki and its later amendments. This study was approved by the local ethics committee of the Medical Faculty at the University of Leipzig (reference number 017/21-ek; date of approval: 19 January 2021).

Cases of acute viral ME in patients who were admitted to the Department of Neurology of the University Hospital Leipzig between January 2015 and December 2020 were reviewed retrospectively. The identification of cases was performed by ICD 10 codes, followed by an in-depth review of hospital records. Cases had to fulfill the following inclusion criteria: 1. Age above 18 years old; 2. Presence of at least one of the following clinical signs suggestive of a viral ME—headache with or without nausea and/or photophobia, epileptic seizure, altered consciousness, a change in personality or behavior, and new focal neurological findings; 3. CSF analysis after lumbar puncture showing a pleocytosis of more than 4 leucocytes per µL; 4. CSF analysis comprising at least leukocytes and CSF/serum ratios for albumin and total immunoglobulin G (IgG), A (IgA), and M (IgM) [9]. Cases with known chronic viral CNS infections (human immunodeficiency virus, JC polyomavirus) were excluded.

Cases were tested for the presence of viral genomes in CSF via real-time (RT)-PCRs, which routinely assessed HSV (in 94.8% of cases), human herpesvirus 6 (in 77.3% of cases), varicella-zoster virus (VZV; in 93.6% of cases), Epstein–Barr virus (in 86.4% of cases), human cytomegalovirus (in 91% of cases), and enteroviruses (in 83.1% of cases). In case of a recent stay in an endemic geographic area (according to the Robert Koch Institute [18]) and corresponding clinical symptoms, additional TBEV testing was performed (in 43.5% of cases). Since 2020, WNV laboratory testing was included, comprising testing for WNV RNA via real-time RT-PCR in CSF, blood and, urine, as well as for WNV-IgG and IgM antibodies via ELISA, followed by virus neutralization assays [6]. Based on the obtained data, West Nile virus cases were identified by following the European Union case definitions for laboratory confirmed and probable WNV infections [19]. The WNV-IgM-positive sera were also tested for TBEV-IgM antibodies (Euroimmun, Lübeck, Germany). The TBEV-IgM ELISA was non-reactive in all of them [6]. Testing for influenza A and B (in 34.0% of cases) was performed during autumn and winter.

All cases were also routinely tested for bacterial ME. For this purpose, direct microscopy was performed, and cultures of the CSF were examined. In addition, the antibody index for *Borrelia burgdorferi* in the serum and CSF was determined. Testing for parasitic or fungal infections, a hematologic disease, or meningeosis was only performed in response to a concrete suspicion (e.g., in case of suspicious findings in the cerebral MRI), in case the patient was immunocompromised, or if a control lumbar puncture revealed an inadequate decrease in the leucocyte count in the CSF despite anti-infective treatment.

In addition, demographic data (age, sex), the season, and clinical data (interval from first symptoms to lumbar puncture, clinical symptoms) were assessed. Cases were further stratified based on the specific virus detections. If no virus was identified, the final diagnosis of a viral CNS infection of unknown etiology was made by a senior physician based on clinical symptoms and course, based on laboratory parameters, and after the exclusion of other infectious or autoimmune causes.

### 2.1. Analysis of the Cerebrospinal Fluid

Measurements of CSF and serum immunoglobulins (G, M, A) and albumin were performed at the Institute of Laboratory Medicine, Clinical Chemistry, and Molecular Diagnostics, University Hospital Leipzig, Leipzig, Germany. Serum immunoglobulins, CSF immunoglobulins, serum albumin, and CSF albumin were analyzed using the Optilite turbidimetric analyzer (The Binding Site Group, Birmingham, UK). The Ig index ((Q_Ig_ = Ig_CSF_*/*Ig_serum_)*/*(Q_Alb_ = Alb_CSF_/Alb_serum_)), the Reiber index ([Q_Ig_ − Q_lim(Ig)_] × Ig_serum_, and the blood–CSF–barrier index damage (Q_Alb_) were calculated. The CSF*/*serum ratios were then plotted into a double-logarithmic diagram. The immunoglobulin-specific graphs in this diagram contained 6 lines: The bold line represented the upper limit of the normal range of the albumin*/*immunoglobulin ratio. The lowest line described the lower limit of the reference range. Generally, elevated levels of immunoglobulins in the CSF may have originated solely from the blood via an impaired blood–CSF–barrier, and*/*or they might be elevated due to an additional intrathecal synthesis, which would be indicative of an inflammation of the central nervous system. In addition, the four dashed lines allowed to estimate the extent (20, 40, 60, and 80%) of intrathecal immunoglobulin synthesis based on the total immunoglobulin content in the CSF. We used freely available software for CSF analysis, i.e., Reiberdiagrams (Albaum IT solutions, Göttingen, Germany), version 4.63.

### 2.2. Statistical Analysis

The software SPSS (version 27.0, IBM Corp., Armonk, NY, USA) was used for statistical calculations. Cases of viral ME were stratified into groups according to the detected virus. After descriptive analyses, statistical significance between groups was assessed via chi-square test for categorical variables and, because of the small sample size, via the Mann–Whitney-U test or Kruskal–Wallis test for metric parameters. Therefore, a *p*-value < 0.05 was defined as statistically significant. To explore whether CSF and demographic parameters may predict the pathogen of the viral CNS infection, a multinominal logistic regression analysis, with the kind of virus (HSV, VZV, WNV, enterovirus, unknown virus) as dependent variables and those variables that differed significantly between groups as factors and co-variables, was calculated.

## 3. Results

From 2015 to 2020, 156 cases of viral CNS infection (74 female, median age 40 years, 25th percentile 29 years, 75th percentile 67.75 years) were treated at a tertiary-care hospital in Germany. Specific virus infections were identified in 93 (59.6%) cases, while in 63 (40.4%) cases, a CNS infection of unknown viral etiology was diagnosed. Cases with a viral CNS infection of known etiology were older (median age of 48 vs. 36 years old, Mann–Whitney-U, *p* = 0.011), while the interval between symptom onset and lumbar puncture was similar between those with known and unknown viral etiologies (Table 1). Laboratory parameters (lymphocyte count, neutrophil count, monocyte count in the CSF, albumin in the CSF, percentage of intrathecal Ig synthesis) did not differ between these two groups (Table 2). Further stratification according to the specific detected virus showed that patients with cases of enterovirus ME were younger (median age 31 years), had a shorter interval from symptom onset to lumbar puncture (2.7 ± 2.6 days), and presented a lesser affection of the blood–CSF–brain barrier (age-adjusted CSF/serum ratio for albumin 10.1 ± 3.3; Table 3) than patients with HSV-, VZV-, or WNV-ME. Cases of WNV-ME were predominantly male (88.9%), older (median age 65 years), had a strong affection of the blood–CSF–brain barrier (age-adjusted CSF/serum ratio for albumin 22.0 ± 6.9), and, in 55.6% of cases, already presented with an intrathecal immunoglobulin M (IgM) synthesis in the initial CSF examination (Table 3 and Figure 1). Moreover, the proportion of neutrophils in the CSF was highest for WNV (48.5 ± 32.6%) and enterovirus (33.6 ± 30.0%) and lowest in VZV (4.8 ± 15.7%) and HSV ME (10.4 ± 20.5). (Table 3). Noteworthily, we found a degree of seasonality, with 37.6% of all viral CNS infections occurring during July to September. Seasonality was highest in WNV (100%) and enteroviruses (40.7%) (Table 3).

A multinominal logistic regression analysis with the kind of virus/unknown virus as the dependent variable; sex, season, intrathecal IgM synthesis as factors; and age and the age-adjusted CSF/serum ratio for albumin as co-variables showed an accuracy for the prediction of HSV of 0%, for VZV of 41.2%, for enteroviruses of 44.4%, for WNV of 88.9%, and for unidentified viruses of 77.4%. Including the percentage of lymphocytes and neutrophils did not increase the accuracy of the prediction of HSV (8.3%), VZV (61.8%), enteroviruses (51.9%), WNV (87.5%), and unidentified viruses (66.7%) significantly.

## 4. Discussion

The present study aimed at identifying patterns of demographic, seasonal, and routine CSF parameters in viral ME that may help to identify the causative pathogen. As a result, in the present cohort, a combination of the following parameters was highly indicative for an infection with WNV: onset of ME in summer, male sex, older age, and prompt evidence of intrathecal total IgM synthesis in the initial CSF sample.

West Nile virus is a mosquito-borne flavivirus. Its natural host reservoir is birds, but the virus can affect a wide range of accidental hosts, including humans. Many mosquitoes are able to spread WNV. *Culex* mosquitoes are among the most efficient ones, which are widespread and highly abundant—not only in Europe. Long known and endemic in large parts of Africa, WNV has subsequently spread globally [20]. In 1999 and the years following, massive outbreaks of WNV infections in humans and animals were reported in Northern America, where the virus quickly established endemicity [21,22,23]. In Europe, so far, highest human case numbers were reported in 2018, and the virus is known to be endemic in the southern, eastern, and central parts of the continent [6,24,25,26]. The geographical spread of WNV is—among others—considered to be associated with specific climatic conditions like unusually warm winters, heat waves during summer, and drought [6,12,27]. It can, therefore, be assumed that the ongoing climate change will promote the further spread of the virus, resulting in a rising number of clinical WNV cases in previously unaffected geographical areas, where knowledge about the virus and its clinical picture is still lacking. Thus, identifying those of the widely assessed routine parameters that might facilitate an early diagnosis of WNV and, in particular, WNV neuroinvasive disease (WNND) is of particular relevance.

The predominance of a white population, male sex, and older age is well known in WNND, which suggests a special vulnerability to neuroinvasive disease in this subset of cases [12,28,29]. However, this pattern is not exclusive to WNND but has also been described in neuroinvasive infections with other members of the genus Flavivirus, like the St. Louis encephalitis virus [30,31] or TBEV [32]. Unlike for many other viruses like HSV, there is a remarkably short interval of only a few days from the start of the WNV viremia to the beginning of the specific antibody synthesis in WNV disease [11]. In general, WNV disease may show not only a monophasic but also a biphasic course similar to TBEV. In particular, the biphasic course with initial flu-like symptoms, followed by an oligosymptomatic phase prior to neurological symptoms due to invasion of the central nervous system, might explain the short latency between first neurological symptoms and antibody synthesis [33,34]. However, in up to 40% of patients with TBE, there is a monophasic course with flu-like symptoms passing over into the symptoms of ME [34].

Noteworthily, like in cases with TBE [32], cases with WNND showed an increased proportion of neutrophils in the differentiation of CSF leucocytes, although the interval from symptom onset to lumbar puncture was significantly longer in WNND than in cases with HSV ME. So far, besides WNND and TBEV, a relevant predominance of a neutrophilic pleocytosis has only been reported for enterovirus ME [17].

This combination of an early intrathecal total IgM synthesis, a neutrophilic pleocytosis, and a severe disruption of the blood–CSF–barrier, as demonstrated by an increased age-adjusted CSF/serum albumin ratio about one week after first symptoms, is even more suggestive of a bacterial than a viral ME, and it might direct the physician’s attention away from the infection with a (Flavi)virus [13,35]. Therefore, we recommend that, particularly during summer, a CSF pattern with an increased intrathecal total IgM synthesis and a neutrophilic pleocytosis should prompt physicians to include Flaviviruses (TBEV and WNV) in their diagnostic workup of ME. Although there is still no established antiviral treatment for TBE and WNND, the timely identification of WNV infections may exclude differential diagnoses and can open up opportunities for timely public health control measures, such as vector control and avoidance or TBE vaccination [12].

Notably, despite a comprehensive diagnostic workup in this study, in about 40% of cases, no specific virus or respective pathogen could be detected, leading to the diagnosis of ME of unknown viral etiology. The overall beneficial clinical course of these patients made a non-viral etiology unlikely. The 40% of cases with ME of unknown viral etiology in this study were in good accordance with larger cohort studies [2]. Recently, the metagenomic analysis of the CSF emerged as a promising approach in ME of unknown viral etiology. Castellot and colleagues found a definite viral cause in about 25% of pediatric ME cases of hitherto unknown viral etiology, involving rare viruses, like human endogenous retrovirus K113, parechovirus 3, and human herpes virus 5, while in another 25% of cases, the results of the metagenomics analysis were not confirmed via PCR [36].

The present study has some limitations. Firstly, due to its retrospective design, it needs confirmation in a larger and prospectively collected cohort. Since viral ME is a rare disease, a multicenter trial should be performed. Secondly, we might have missed cases of WNND prior to the 2020 outbreak, since WNV testing had not been performed in all cases beforehand [6]. As the first (low-level) autochthonous transmission of WNV in resident birds and horses in the area had already been diagnosed in 2018 [37], it is possible that human WNND cases were missed in 2018 or 2019 and attributed to the group of ME cases with unknown viral etiology.

## Figures and Tables

**Figure 1 viruses-16-00341-f001:**
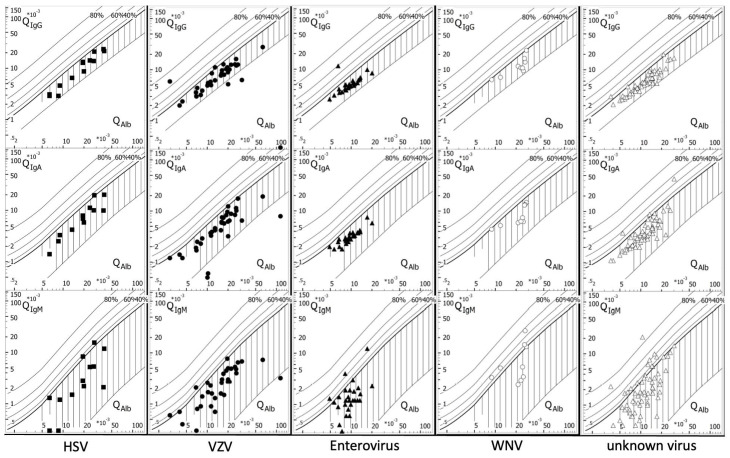
Intrathecal synthesis of immunoglobulins G, M, and A (Reiber diagram) in viral meningitis/encephalitis (ME) via herpes simplex virus (HSV), varicella-zoster virus (VZV), enterovirus, and West Nile virus (WNV) and in ME of unknown viral etiology. Genomes of the specific viruses were detected in the cerebrospinal fluid (CSF) via real-time polymerase chain reaction. The diagnosis of a ME of unknown viral etiology was made after an extensive diagnostic workup without evidence for a specific pathogen or autoimmune cause. The x-axis represents the quotient (Q) of albumin (Alb) in the CSF and the serum (Q_Alb_ = Alb_CSF_/Alb_serum_), while the y-axis shows the quotient of the immunoglobulin (G, A, M) concentration in the CSF and the serum (Q_Ig_ = Ig_CSF_/Ig_serum_). The ratios of Q_Alb_ and Q_Ig_ were plotted into the diagram. The bold printed curve stands for the upper reference range and represents the discrimination line between brain-derived and blood-derived immunoglobulin fractions in the CSF.

**Table 1 viruses-16-00341-t001:** Baseline and clinical data of cases with viral meningitis/encephalitis. * Mann–Whitney-U test, ^#^ chi square test.

	All Cases (*n* = 156)	Cases with Identified Virus (*n* = 93)	Cases with Unknown Viral Etiology (*n* = 63)	Statistical Significance
Age (in years, median, 25th percentile, 75th percentile)	40 (29, 67.75)	48 (31.5, 73.5)	36 (28, 60)	0.011 *
Sex (female, %)	74 (47.4%)	42 (45.2%)	32 (50.8%)	0.489 ^#^
Season (*n*, %)				0.115 ^#^
spring	33 (21.2%)	19 (20.4%)	14 (22.2%)	
summer	58 (37.2%)	37 (39.8%)	21 (33.3%)	
autumn	36 (23.1%)	25 (26.9%)	11 (17.5%)	
winter	29 (18.6%)	12 (12.9%)	17 (27.0%)	
Time from symptom onset to lumbar puncture (in days)	4.9 ± 4.5	4.7 ± 4.4	5.1 ± 4.7	0.765 *

**Table 2 viruses-16-00341-t002:** Comparison of routine cerebrospinal fluid (CSF) parameters between cases with a known virus and these without the detection of a virus. *Ig* immunoglobulin. * Mann–Whitney-U test, ^#^ chi square test.

	All Cases (*n* = 156)	Cases with Identified Virus (*n* = 93)	Cases with Unknown Viral Etiology (*n* = 63)	Statistical Significance
Leucocytes CSF (/µL)	203 ± 311	225 ± 355	169 ± 228	0.185 *
Lymphocytes CSF (%)	58.3 ± 30.1	58.3 ± 30.1	60.0 ± 31.2	0.627 *
Neutrophils CSF (%)	21.1 ± 28.2	19.3 ± 28.1	23.6 ± 28.5	0.163 *
Monocytes CSF (%)	16.6 ± 15.3	18.4 ± 15.7	14.0 ± 14.5	0.073 *
Albumin CSF (mg/L)	763 ± 468	834 ± 547	660 ± 299	0.060 *
Age-adjusted CSF/serum ratio for albumin	14.0 ± 8.4	15.4 ± 9.6	11.9 ± 5.7	0.032 *
Intrathecal Ig synthesis	29 /153 (19.0%)	17/91 (18.7%)	12/62 (19.4%)	0.917 ^#^
Intrathecal IgM synthesis	22 /153 (14.4%)	12/91 (13.2%)	10/62 (16.1%)	0.611 ^#^
Intrathecal IgG synthesis	7/153 (4.6%)	5/91 (5.5%)	2/62 (3.2%)	0.510 ^#^
Intrathecal IgA synthesis	5/153 (3.3%)	3/91 (3.3%)	2/62 (3.2%)	0.981 ^#^

**Table 3 viruses-16-00341-t003:** Comparison of routine cerebrospinal fluid (CSF) parameters in cases with meningitis/encephalitis (ME) depending on the virus. Ig, immunoglobulin; HSV, herpes simplex virus; VZV, varicella-zoster virus; WNV, West Nile virus. * Kruskal–Wallis test as a global test for comparisons between parameters; ^#^ chi square test.

	Cases of HSV-ME (*n* = 14)	Cases of VZV-ME (*n* = 36)	Cases of Enterovirus-ME (*n* = 27)	Cases of WNV-ME (*n* = 9)	Cases of ME of Unknown Viral Etiology (*n* = 63)	Statistical Significance
Age (median, in years)	67.5	66	31	65	36	<0.001 *
Sex (female, %)	6 (42.9%)	23 (63.9%)	9 (33.3%)	1 (11.1%)	32 (50.8%)	0.024 ^#^
Season (*n*, %)						0.003 ^#^
spring	1 (7.1%)	6 (16.7%)	9 (33.3%)	0	14 (21.9%)	
summer	5 (35.7%)	12 (33.3%)	11 (40.7%)	9 (100%)	22 (34.4%)	
autumn	6 (42.9%)	12 (33.3%)	6 (22.2%)	0	11 (17.2%)	
winter	2 (14.3%)	6 (16.7%)	1 (3.7%)	0	17 (26.6%)	
Time from symptom onset to lumbar puncture (in days)	3.5 ± 3.2	6.1 ± 5.3	2.7 ± 2.6	6.6 ± 4.0	5.1 ± 4.7	0.007 *
Leucocytes CSF (×10^6^/L)	228 ± 449	224 ± 258	176 ± 160	147 ± 157	169 ± 228	0.270 *
Lymphocytes CSF (%)	60.9 ± 34.1	72.0 ± 24.4	44.6 ± 26.9	29.5 ± 20.8	60.0 ± 31.2	<0.001 *
Neutrophils CSF (%)	10.4 ± 20.5	4.8 ± 15.7	33.6 ± 30.0	48.5 ± 32.6	23.6 ± 28.5	<0.001 *
Monocytes CSF (%)	25.6 ± 18.8	16.6 ± 14.3	19.2 ± 16.7	17.2 ± 13.4	14.0 ± 14.5	0.230 *
Albumin CSF (mg/L)	811 ± 401	992 ± 693	564 ± 180	909 ± 267	660 ± 299	<0.001 *
Age-adjusted CSF/serum ratio for albumin	19.3 ± 11.5	15.9 ± 10.5	10.1 ± 3.3	22.0 ± 6.9	11.9 ± 5.7	<0.001 *
Intrathecal Ig synthesis (*n* = 155, %)	2/14 (14.3%)	3/34 (8.8%)	5/27 (18.5%)	5/9 (55.6%)	12/62 (19.4%)	0.033 ^#^
Intrathecal IgM synthesis (*n* = 155, %)	1/14 (7.1%)	2/34 (5.9%)	4/27 (14.8%)	5/9 (55.6%)	10/62 (16.1%)	0.006 ^#^

## Data Availability

All data generated or analyzed during this study are included in this published article and are available from the corresponding author upon reasonable request.

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
