# Peer review of "A Specific Pattern of Routine Cerebrospinal Fluid Parameters Might Help to Identify Cases of West Nile Virus Neuroinvasive Disease"

_viruses, 2024, doi:10.3390/v16030341_

Round 1

Reviewer 1 Report

Comments and Suggestions for Authors

I suggest minor revision.

Reviewer 2 Report

Comments and Suggestions for Authors

The brief report "A specific pattern of routine cerebrospinal fluid parameters might help to identify cases of West Nile virus neuroinvasive disease" describes a retrospective review of clinical data of cases of meningitis/encephalitis with the purpose of highlighting parameters indicative of human infection with WNV. Overall the study is presented clearly but needs to a address a number of points around the clear description of what has been done, how, and what is significant. The following should be addressed:

1. The authors refer to the Reiber diagram at a number of points, arguably inappropriately, without adequate explanation of what this is and its significance. Firstly, reference to this in the abstract should be removed and limited to "and intrathecal total immunoglobulin M synthesis detected in the first CSF sample." Intrathecal IgM was not synthesised in the Reiber diagram. It would be helpful for the authors to explain why albumin and antibody levels in the CSF are diagnostically useful and that plotting on a Reiber diagram helps with clinical interpretation, specifically the status of the blood-brain-barrier. There is no mention in the Methods section on how Albumin, IgA, IgM & IgG levels were measured, how they were plotted to create the Reiber diagram or the diagram interpreted. This must be included. Some explanation on why cases are assumed to be of viral etiology and justify why the term "unknown virus" can be used in Figure 1.

2. The Results section must be improved/revised. Each Table contains a "p value" column, this should be " statistical significance". It is also not clear what comparison the values refer to as the Tables include multiple parameters. Even worse, not a single statistical value is included in the text so it is unclear which of the parameters cited is supported statistically.

Reviewer 3 Report

Comments and Suggestions for Authors Dear authors, Your manuscript describes the evaluation of involvement of different viruses in viral meningitis/encephalitis (ME) diseases, especially West-Nile virus and analysis of demographic, seasonal, and routine cerebrospinal fluid (CSF) parameters in cases of viral ME to assess their utility for the prediction of the causative virus. Such approach if very important for fast diagnostics which is still missing in some cases because the nature of ME causative agent plays the important role in treatment strategies which differ in case of different viruses. One of the undoubted advantages of this work is that it covers a wide range of infectious agents, which is often not done in other studies. The manuscript is really well written, contains all necessary methodology, the conclusions are consistent with the evidence and arguments presented in the discussion section. The references are appropriate, and tables and figures provide enough and adequate data. I have only few issues regarding it: General remark: based on the manuscript’s title it should be expected that the study will focus only on WNV, but you also tested MEs caused by other viruses. My suggestion is to change the title of the manuscript. L87 – “REF RKI” – what is it? Specify, please.

Round 2

Reviewer 2 Report

Comments and Suggestions for Authors

All reviewers comments have been addressed.